

# *Kaempferia chonburiensis* (Zingiberaceae), a new species from Thailand based on morphological and molecular evidence

Pornpimon Wongsuwan[1,2], Boonmee Phokham[3], Pantamith Rattanakrajang[1], Chayan Picheansoonthon[4] and Suchada Sukrong[1,5]

[1] Center of Excellence in DNA Barcoding of Thai Medicinal Plants, Department of Pharmacognosy and Pharmaceutical Botany, Faculty of Pharmaceutical Sciences, Chulalongkorn University, Bangkok, Thailand
[2] Division of Applied Thai Traditional Medicine, Faculty of Medicine, Mahasarakham University, Maha Sarakham, Thailand
[3] Abhaibhubejhr College of Thai Traditional Medicine Prachinburi, Faculty of Public Health and Allied Health Sciences, Praborommarajchanok Institute, Prachin Buri, Thailand
[4] Academy of Science, The Royal Society (Thailand), Bangkok, Thailand
[5] Chulalongkorn School of Integrated Innovation, Chulalongkorn University, Bangkok, Thailand

Corresponding author
Suchada Sukrong,
suchada.su@chula.ac.th

## ABSTRACT

**Background:** *Kaempferia* is a genus belonging to the ginger family. Currently, this genus is comprised of about 63 species, mainly distributed from India to Southeast Asia. During our fieldwork, a new species of *Kaempferia* was found in Chon Buri Province, Thailand. The objective of this article was to provide morphological evidence and confirm its relationships in *Kaempferia* through molecular phylogenetic analysis.
**Methods:** Plant samples were collected from field sites and investigated by conventional taxonomy and molecular techniques. The phylogenetic trees were reconstructed using the maximum likelihood criterion and Bayesian inference. The morphological evolution was also examined to elaborate the relationships among representative *Kaempferia* taxa.
**Results:** *Kaempferia chonburiensis* from southeastern Thailand is described and illustrated based on morphological features and its taxonomic placement was confirmed by molecular phylogenetic analyses and morphological evolution. An identification key is provided for the new *Kaempferia* species occurring in Thailand.
**Conclusion:** *Kaempferia chonburiensis* is a new enumerated species of *Kaempferia* from Thailand.

## INTRODUCTION

The genus *Kaempferia* L. encompasses several species of notable ethnomedicinal value, particularly within traditional and indigenous medicine contexts (*Picheansoonthon & Koonterm, 2008*; *Pham, Nguyen & Nguyen, 2021*). This genus, belonging to the tribe Zingibereae of the ginger family (Zingiberaceae), was established in 1753 with the initial classification of two species: *K. galanga* L. and *K. rotunda* L. (*Linnaeus, 1753*). *Kaempferia*

taxa are categorized into two subgenera based on the appearance of their inflorescences: *K.* subgenus *Kaempferia* and subgenus *Protanthium* (Horan.) Baker (*Horaninow, 1862*; *Baker, 1890*; *Kam, 1980*; *Insisiengmay, Newman & Haevermans, 2018*). The former produces inflorescences at the tips of pseudostems (*Insisiengmay, Newman & Haevermans, 2018*), while the latter produces inflorescences directly from the rhizome before the leafy shoot arises (*Insisiengmay, Newman & Haevermans, 2018*; *Noppporncharoenkul et al., 2021*). The genus is distributed from India across Southeast Asia (*Kress, Prince & Williams, 2002*; *Larsen & Larsen, 2006*; *POWO, 2024*), with Thailand recognized as one of the hotpots of its diversity (*Larsen & Larsen, 2006*; *Leong-Škorničkova & Newman, 2015*). The genus presently comprises approximately 63 accepted species names (*POWO, 2024*). Over the past two decades, a substantial number of new *Kaempferia* taxa have been described and identified in Thailand. Currently, 32 species are reported in the Flora of Thailand, with 20 species belonging to subgenus *Kaempferia* and 12 species to *Protanthium* (*Jenjittikul, Noppporncharoenkul & Ruchisansakun, 2023*). The most recent additions to the subgenus *Kaempferia* include *K. pseudoparviflora* Saensouk & P. Saensouk (*Saensouk & Saensouk, 2021a*), *K. isanensis* Saensouk & P. Saensouk, *K. unifolia* Saensouk & P. Saensouk (*Saensouk & Saensouk, 2021b*), *K. nigrifolia* Boonma & Saensouk (*Boonma, Saensouk & Saensouk, 2021*), *K. sakonensis* Saensouk, P. Saensouk & Boonma, *K. napavarniae* Saensouk, P. Saensouk & Boonma (*Saensouk, Saensouk & Boonma, 2022*), and *K. sakolchaii* P. Saensouk, Saensouk & Boonma (*Saensouk et al., 2024*). Additionally, the latest additions to the subgenus *Protanthium* are *K. sipraiana* Boonma & Saensouk (*Boonma, Saensouk & Saensouk, 2022*), *K. subglobosa* Noppornch. & Jenjitt (*Noppporncharoenkul & Jenjittikul, 2024*), and *K. calcicola* Noppornch. (*Noppporncharoenkul et al., 2024*). To date, about 42 species, including approximately 20 endemic species, are enumerated in Thailand.

Given the effectiveness of molecular methods in determining phylogenetic relationships, as well as in identifying and discriminating medicinal plants, this study focused on specific DNA regions that distinguish among various medicinal plant species. Despite *Kaempferia*'s ethnobotanical value, economic importance, and species diversity, there has been limited scientific investigations on its molecular characterization. DNA sequences of the internal transcribed spacer (ITS) and *maturase* K (*mat*K) regions have been successfully used for the infrafamilial classification and phylogenetic studies within the family Zingiberaceae (*Kress, Prince & Williams, 2002*). For Thai *Kaempferia* species, the genetic diversity and species identification have also been explored using maternally inherited chloroplast *psb*A-*trn*H and partial *pet*A-*psb*J spacer sequences (*Techaprasan et al., 2010*), as well as a nuclear internal transcribed spacer, ITS2 (*Noppporncharoenkul et al., 2016*). The ITS region has also been used to identify ethnomedicinally important *Kaempferia* species in Malaysia (*Labrooy, Abdullah & Stanslas, 2018*) and Vietnam (*Van et al., 2021*) along with *trn*L-F sequences. Additionally, the ITS and *mat*K have been used to differentiate medicinal *Kaempferia* species from their adulterants (*Basak et al., 2019*). However, there is limited information available on the phylogenetic relationships of *Kaempferia* species and many insufficiently characterized data.

Several members of *Kaempferia* are still taxonomically unidentified, and various taxa are currently under thorough investigation for drug development purposes. Our field surveys on the medicinal plants of Zingiberaceae in Thailand resulted in the discovery of a new species of *Kaempferia*. The analyses based on morphological characteristics indicate its difference from other morphologically similar species previously described in the genus *Kaempferia*. Additionally, phylogenetic reconstruction was performed using nuclear and chloroplast DNA regions that also supported this result. The objective of this study was to describe this species as new, and to provide a key for identification of *Kaempferia* from Thailand.

## MATERIALS AND METHODS

### Morphological and taxonomical studies

The morphological characteristics of vegetative and reproductive structures were studied using fresh and preserved specimens collected from the public natural habitat in Chon Buri Province, southeastern Thailand, spanning from 2021 to 2023 (Fig. 1). Various characters such as rhizomes, leaves, bracts, peduncles, labellum, staminodes, and ovaries were carefully examined, measured, and photographed. These characteristics were compared with information published in various literature sources (*Picheansoonthon & Koonterm, 2008*; *Saensouk, Saensouk & Boonma, 2022*; *Jenjittikul, Nopporncharoenkul & Ruchisansakun, 2023*; *Nopporncharoenkul et al., 2024*), as well as with allied species deposited in several herbaria at Bangkok Herbarium (BK), Bangkok Forest Herbarium (BKF), Queen Sirikit Botanic Garden Herbarium (QBG), and Singapore Herbarium (SING). Additionally, available online information and digital images of herbarium specimens, the Kew Herbarium Catalogue (https://apps.kew.org/herbcat/navigator.do) were considered, along with existing published literature on *Kaempferia*, which was also compared in this study. The most morphologically similar species were compared in Table 1. All the authentic specimens of the plants were dried, and herbarium specimens were prepared following standard procedures (*de Almeida et al., 2023*) before being deposited in BKF. A distribution map was created using the SimpleMappr online application (*Shorthouse, 2010*).

The electronic version of this article in Portable Document Format (PDF) will represent a published work according to the International Code of Nomenclature for algae, fungi, and plants (ICN), and hence the new names contained in the electronic version are effectively published under that Code from the electronic edition alone. In addition, new names contained in this work which have been issued with identifiers by IPNI will eventually be made available to the Global Names Index. The IPNI LSIDs can be resolved and the associated information viewed through any standard web browser by appending the LSID contained in this publication to the prefix "http://ipni.org/". The online version of this work is archived and available from the following digital repositories: PeerJ, PubMed Central SCIE, and CLOCKSS.

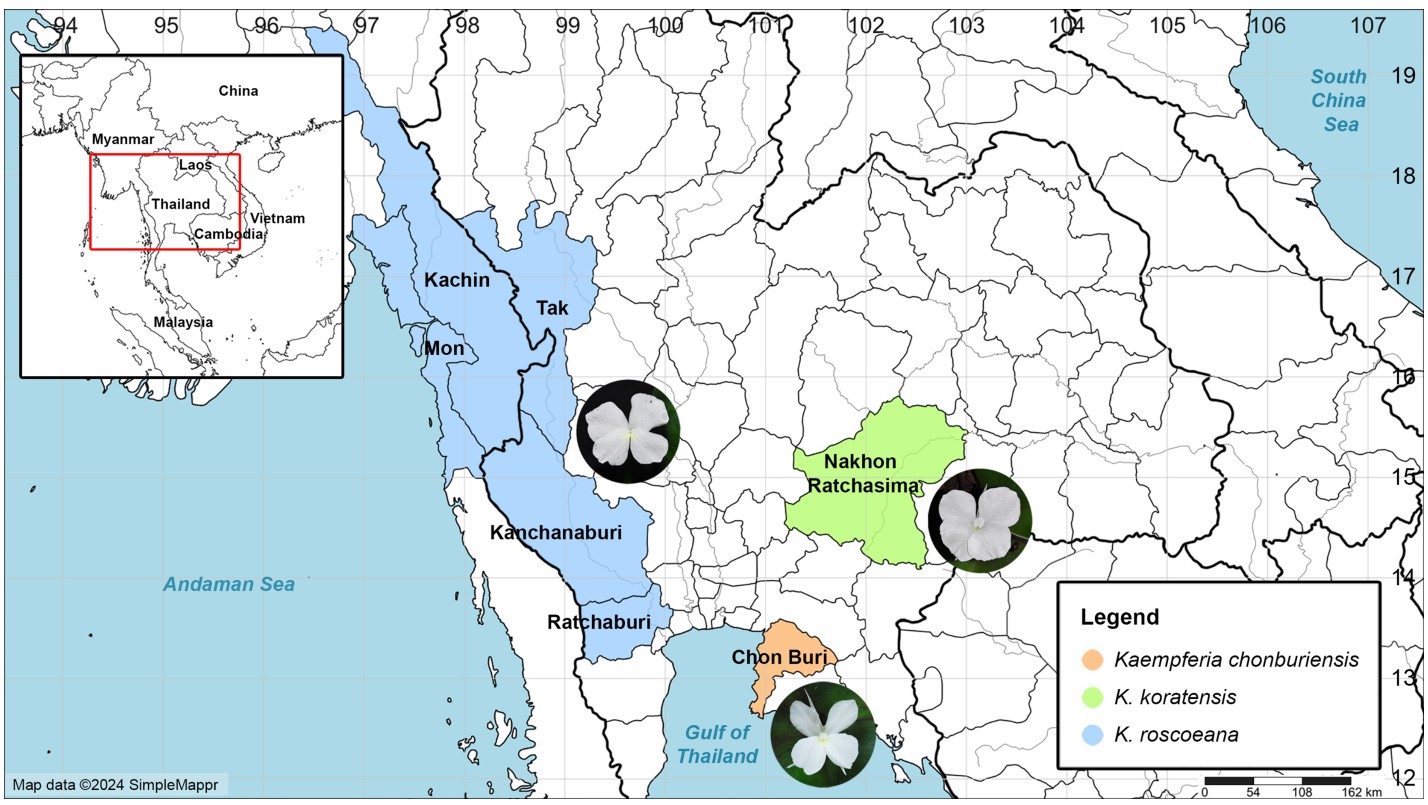

**Figure 1 Map of the distribution of *Kaempferia chonburiensis* (orange) and two morphologically similar species, *K. koratensis* (green) and *K. roscoeana* (blue).** Map data © 2024 SimpleMappr (prepared by Pantamith Rattanakrajang and photos by Pornpimon Wongsuwan).

**Table 1 Comparison of morphological characters and geographic data among *K. chonburiensis* and its closely related species.**

| | K. chonburiensis | K. koratensis | K. galanga | K. roscoeana |
|---|---|---|---|---|
| **1. Geographic distribution** | Endemic to Thailand (Chon Buri) | Endemic to Thailand (Nakhon Ratchasima) | Widespread: Bangladesh, Cambodia, China, India, Indonesia, Laos, Malaysia, Myanmar, Singapore, Thailand, Vietnam | Myanmar, Thailand (Kanchanaburi, Ratchaburi, Tak) |
| **2. Elevation** | 45–50 m | 224–350 m | Up to 1,500 m | N/A |
| **3. Leaf** | | | | |
| Ligule | 0.6–1.0 cm long, pubescent | ca. 3 mm long, pubescent | 1–3 mm long, pubescent | ca. 2 mm long, glabrous |
| Upper surface | Glabrous | Glabrous | Glabrous | Glabrous |
| Lower surface | Pubescent | Pubescent | Pubescent | Glabrous |
| **4. Inflorescence** | | | | |
| Peduncle | 0.8–1.2 cm long | ca. 5 mm long | Sessile | Sessile |
| **5. Flower** | | | | |
| Bract | Lanceolate to oblong, 3.1–4.0 × 0.9–1.7 cm, glabrous | Oblong, 2.4–4.3 × 0.6–2.0 cm, sparsely pubescent | Lanceolate to oblong, 2.9–4.1 × 0.9–1.7 cm, glabrous | Oblong, glabrous, 2.0–2.2 cm × 3–4 mm, glabrous |

| | K. chonburiensis | K. koratensis | K. galanga | K. roscoeana |
|---|---|---|---|---|
| Labellum | Broadly obovate to suborbicular, each lobe obovate to broadly obovate, 2.6–3.1 × 1.5–2.1 cm, apex undulated, rounded, white with yellow spot at the base, with or without violet patch | Broadly obovate to suborbicular, each lobe broadly obovate or suborbicular, 2.2–2.7 × 1.8–2.5 cm, apex rounded, emarginated, white with yellow spot at the base | Broadly obovate to suborbicular, each lobe obovate to broadly obovate, 1.9–2.7 × 1.2–2.1 cm, apex undulated, rounded, white with yellow spot at the base, with violet patch | Obovate, each lobe obovate, 1.4–2.1 × 1.1–1.5 cm, white with yellow spot at the base, apex rounded |
| Anther-crest | Obovate or rectangular, 4–6 × 3–4 mm, apex rounded, bilobed | Ovate to obovate or rectangular, 0.7–1.2 cm × 4–7 mm, apex rounded, bi-to-trilobed | Obovate, 4–5 × 3–4 mm, apex rounded, shallowly bilobed | Ovate, ca. 1.5 × 1.5 mm, apex acute |
| Ovary | Pubescent at the uppermost part | Glabrous | Glabrous | Glabrous |

## Taxon sampling, DNA extraction, amplification, and sequencing

The phylogenetic placement of the new species was discovered based on an ingroup sampling comprising 17 accessions of 15 species of *Kaempferia*, including three individuals of the new species from the type locality in Chon Buri Province, Thailand (Table S1). *Zingiber wrayi* was retrieved from the GenBank database to complement our dataset as an outgroup (Table S1). For our molecular analysis, one nuclear DNA marker, the ribosomal internal transcribed spacers (ITS), and one chloroplast DNA marker, *maturase* K (*mat*K), were chosen. A total of 34 new sequences were executed from our study and submitted to the GenBank database. Voucher information and GenBank accession numbers for all sequences are listed in Table S1.

Approximately 100 mg of fresh leaves from each *Kaempferia* sample were used for DNA extraction, utilizing the DNeasy Plant Mini Kit (QIAGEN, Germany) as per the manufacturer's protocol. The concentration of the extracted DNA was measured with a NanoDrop™ Microvolume UV-Vis Spectrophotometer (Thermo Fisher Scientific, Waltham, MA, USA).

The isolated genomic DNA served as the template for amplifying the nuclear ITS and chloroplast *mat*K regions, using the universal primers. For the ITS regions, the primers used ITS1 (5′- TCC GTA GGT GAA CCT GCG G-3′), ITS4 (5′-TCC TCC GCT TAT TGA TAT GC-3′), and ITS5 (5′- GGA AGT AAA AGT CGT AAC AAG-3′) according to *White et al. (1990)*. The primers *trn*K-3914F (5′-TGGGTTGCTAACTCAATG-3′) and *trn*K-2R (5′-AACTAGTCGGATGGAGTAG-3′) were used for *mat*K regions (*Johnson & Soltis, 1994*). Each 25 μl polymerase chain reaction (PCR) reaction contained 1X Platinum™ II PCR Buffer, 0.4 mM dNTPs, 0.5 μM primers, 0.04 U/μl Platinum™ II Taq Hot-Start DNA Polymerase, and 1 μl of DNA template. PCR reactions were conducted under the following conditions: initial denaturation at 94 °C for 4 min, followed by 30 cycles of denaturation at 94 °C for 30 s, annealing at 55 °C for ITS or 60 °C for *mat*K for 30 s, and extension at 72 °C for 45 s for ITS or 1 min 30 s for *mat*K, using a GS-96 Gradient Touch Thermal Cycler (Hercuvan, Selangor, UK). After PCR amplification, the PCR amplicons were examined on a 1.5% (w/v) agarose gel in 1× TAE buffer containing 1× RedSafe nucleic acid staining

solution. The amplified PCR products were sequenced using Sanger sequencing (Barcode-Tagged Sequencing; CELEMICS, Seoul, South Korea).

## Phylogenetic analysis

The phylogenetic analysis of *Kaempferia* DNA sequencing data was conducted using the CIPRES Science Gateway (https://www.phylo.org/). The accession numbers of reference sources and *Kaempferia* samples were provided in Table S1. All the sequences were aligned using MAFFT on XSEDE (*Katoh et al., 2002*). The selected substitution model of the concatenated data (ITS + *mat*K) was general time reversible (GTR)+I+Γ with the number of substitutions set at six, determined using Jmodeltest2 on XSEDE (2.1.10) (*Darriba et al., 2012*). Maximum likelihood (ML) was performed using IQ-Tree (XSEDE, on ACCESS v.2.2.2.7) with 1,000 ultra-fast bootstrap replicates (*Nguyen et al., 2015*). For Bayesian Inference (BI), the analysis of the concatenated dataset was executed using MrBayes v.3.2.6 (*Ronquist et al., 2012*) on XSEDE (*Towns et al., 2014*). Six substitution types were computed in matrices with the proper model, GTR+I+Γ. Two simultaneous Markov Chain Monte Carlo (MCMC) methods were calculated for ten million generations with sampling every 250 generations, with discarding the initial 25% burn-in of the sampled trees. The ML concatenated dataset was visualized as a representative phylogenetic tree with bootstrap support (BS) and posterior probability (PP), using the Figtree v.1.4.3 program (*Rambaut, 2016*).

## Character state analysis

Seven characters based on morphology and floristic geographic regions of Thailand were studied as shown in Table 2. The characters were coded based on our observation of living specimens, photographs, and published literatures (*Picheansoonthon & Koonterm, 2008*; *Saensouk, Saensouk & Boonma, 2022*; *Jenjittikul, Nopporncharoenkul & Ruchisansakun, 2023*; *Nopporncharoenkul et al., 2024*), summarized in Table S2. All the character states were combined with the representative phylogenetic tree to reconstruct a character state tree using packages phytools (*Revell, 2024*) and treeio (*Wang et al., 2020*) executed in R programming.

# RESULTS

## Taxonomic study

The morphological characters, especially ligules and flowers, of the new species differ from all species previously described in *Kaempferia* and belong to the subgenus *Kaempferia*. The undescribed *Kaempferia* species can be distinguished from the most morphologically similar species, *K. koratensis*, by the following diagnostic characters: a longer ligule, a longer peduncle, glabrous bracts, white flowers with a yellow spot at the base, as well as a labellum that may or may not have a violet patch, and pubescent ovary at the uppermost part.

## Molecular phylogenetic analysis

The molecular phylogenetic relationships among *Kaempferia* species are indicated in the phylogenetic tree based on the combined dataset from ITS and *mat*K (Fig. 2). The

**Table 2 The morphological characteristics and floristic geographic regions of Thailand used to character state analysis in this study of *Kaempferia*.**

|  | Character states |
| --- | --- |
| Geography | East/North/Northeast/Peninsular/Southeast/Southwest |
| Rhizome | Moniliform/not moniliform |
| Ligule length | Absent - 5 mm/>5 mm |
| Leaves orientation | Horizontal/not horizontal |
| Inflorescence | Before leaves/with leaves |
| Peduncle length | >5 mm/sessile - 5 mm |
| Staminodes and labellum | Not same plane/same plane/staminodes absent |

concatenated nuclear and chloroplast datasets exhibited the best-fitting model for the phylogenetic tree reconstruction in this study, compared to each individual dataset, including the congruence between both two datasets. The topologies of ML and BI trees are almost identical, and the bootstrap support values and posterior probability are also provided in Fig. 2. Using *Zingiber wrayi* to root the tree, the monophyly of *Kaempferia* was strongly supported as revealed by the maximum bootstrap (BS = 100, PP = 1). There are three major clades found in the tree as follows: (1) clade A (BS = 91, PP < 0.80) comprises four minor clades–clade A1 (BS = 91, PP = 1) includes *K. chonburiensis*, *K. galanga* and *K. koratensis*; clade A2 (BS < 80, PP = 0.80) includes *K. larsenii*, *K. marginata*, and *K. sisaketensis*; clade A3 (BS = 85, PP < 0.80) includes *K. pulchra* and *K. roscoeana*, and clade A4 (BS = 99, PP = 0.80) includes *K. elegans* and *K. parviflora*; (2) clade B (BS = 99, PP = 1) includes *K. angustifolia*, *K. fissa*, and *K. minuta*; and (3) clade C (BS = 97, PP = 1) includes *K. rotunda* and *K. udonensis*. In the major clades, the subgenus *Kaempferia* (clades A-B) is paraphyletic with the exclusion of monophyletic subgenus *Protanthium* (clade C). The position of three accessions of an undescribed species (labelled as *K. chonburiensis*) in clade A is grouped together (BS = 87, PP = 1) having *K. koratensis* and *K. galanga* as its sister group and closely related species.

### Morphological evolution in *Kaempferia*

The morphological characters of the *Kaempferia* species in our study exhibit homoplasy, except the appearance of inflorescences (Fig. 3). Three major clades in the tree are observed as same as the previous phylogenetic tree (Fig. 2). We determined that the key character of *Kaempferia* subgenus *Kaempferia* (clades A and B) is having inflorescence appearing with leaves. In *Kaempferia* subgenus *Protanthium* (clade C), the key character is the inflorescence appearing before leaves. In addition, the floristic geographic regions of Thailand of each species were variable based on the site where the plants were collected.

## DISCUSSION

### Morphological characteristics

At first glance, the undescribed species (labeled as *K. chonburiensis*) morphologically resembles *K. koratensis* circumscribed by *Picheansoonthon (2011)* in leaf shape and flower

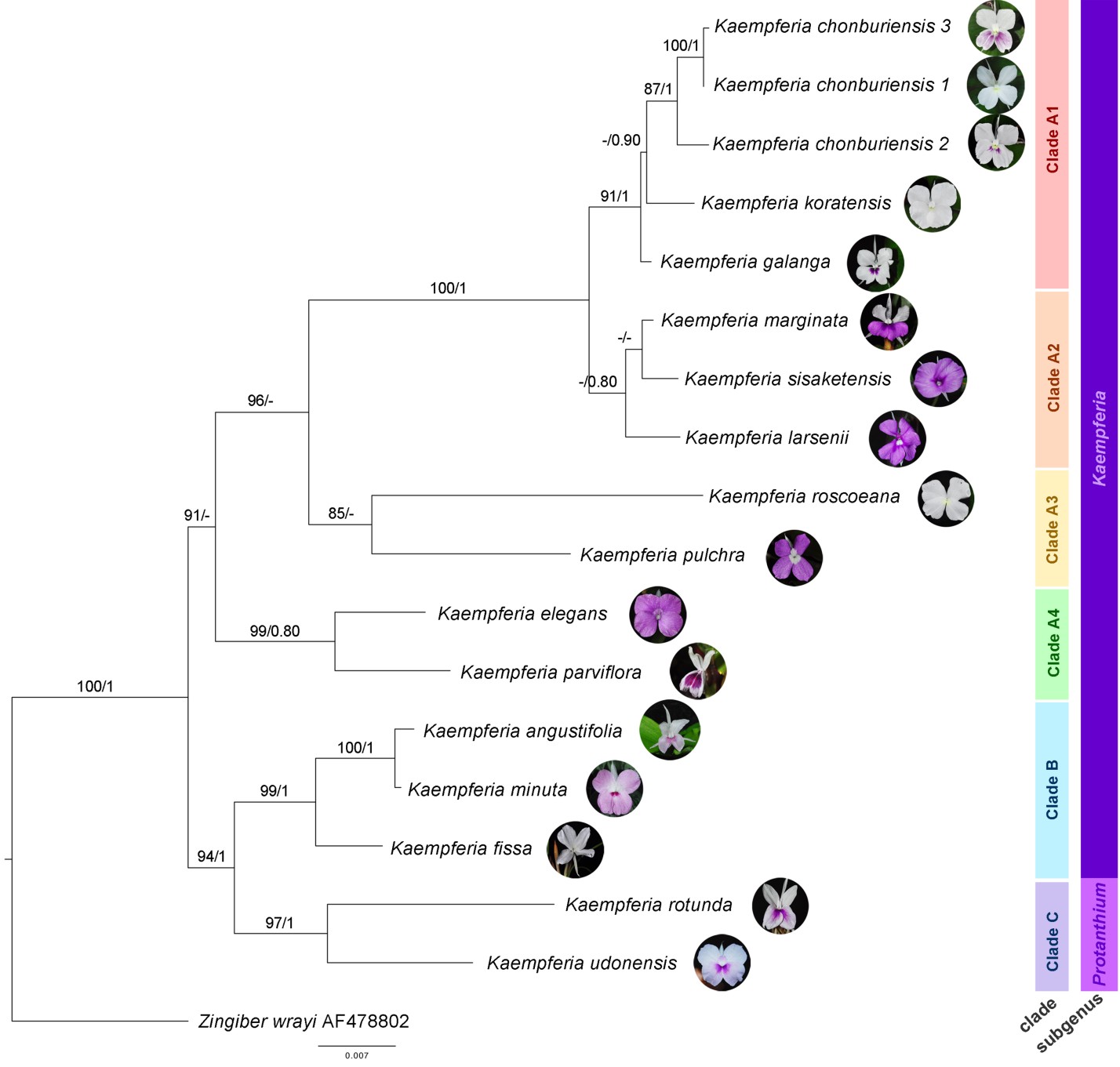

**Figure 2 Phylogram depicting the maximum likelihood IQ-Tree based on the concatenated dataset (ITS and *mat*K) for the genus *Kaempferia*, under the GTR+I+Γ reversible evolution model.** All branches with maximum likelihood (ML) bootstrap percent support. Each clade is separated based on morphological and molecular evidence. Two subgenera were traditionally described by *Horaninow (1862)* and *Baker (1890)*. The tree is rooted by an outgroup species, *Zingiber wrayi*. ML bootstraps (BS ≥ 80%) and BI posterior probabilities (PP ≥ 0.80) are presented above branches (prepared by Pantamith Rattanakrajang and photos by Pornpimon Wongsuwan).

color. However, after intensive morphological analysis and comparisons (Table 1; Fig. 3), it became clear that this undescribed species could not be taxonomically identified as any of the existing species within subgenus *Kaempferia*. The undescribed species can be obviously

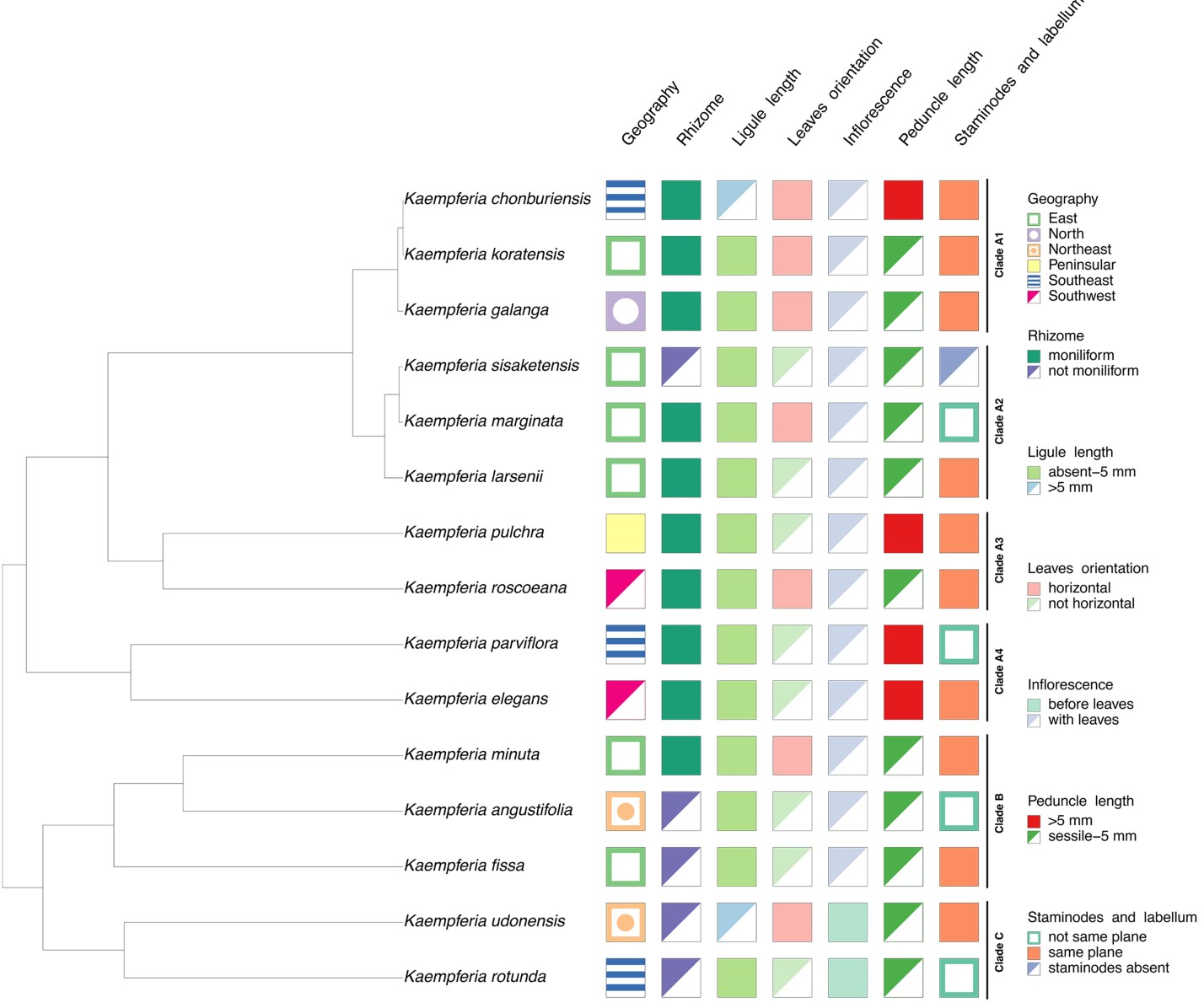

**Figure 3 Character tree of the genus *Kaempferia*.** Seven morphological character states traced with the representative phylogenetic tree display the morphological evolution among *Kaempferia* species. All the clades were based on the phylogram in Fig. 2.

distinguished from *K. koratensis* by several consistent characters, including a longer ligule (usually > 6 mm), a longer peduncle (usually > 8 mm), a glabrous bract, white with a yellow spot at the base, as well as with or without a violet patch on the labellum, and the ovary is pubescent at the uppermost part (*Picheansoonthon, 2011*). These characteristics of the undescribed species could be substantially indicated as autapomorphies.

Moreover, the color variation from white-to-violet in the patch on the labellum was observed in the new species. Likewise, the white-to-violet patch on the labellum in some flowers in *Kaempferia*, *K. galanga* and *K. marginata* suggests the potential intraspecific

diversity in the cryptic species complexes or natural hybrid species (*Gangmei et al., 2024*). Further investigation could explore whether color differences correspond to other ecological, genetic, or reproductive differences, such as suitable niches or behavior differences. This could provide insights into the evolutionary processes within *Kaempferia* and help improve understanding of the complex nature of flower traits in this group.

## Phylogenetic relationships and morphological evolution within the genus *Kaempferia*

Our resolved backbone of the phylogeny of *Kaempferia* supports that the genus *Kaempferia* is monophyletic, as being consistent with other previous phylogenetic studies (nuclear ITS and chloroplast *mat*K in *Kress, Prince & Williams (2002)*; chloroplast *psb*A-*trn*H and *pet*A-*psb*J spacers in *Techaprasan et al. (2010)*; nuclear ITS2 in *Nopporncharoenkul et al. (2016)*).

In the traditional infrageneric classification, the morphological distinction of two subgenera had been supported by the appearance of inflorescences on the leafy shoot (subgenus *Kaempferia*) and on the separate shoot before leaves (subgenus *Protanthium*) (*Horaninow, 1862*; *Baker, 1890*; *Kam, 1980*; *Insisiengmay, Newman & Haevermans, 2018*). However, the paraphyly of subgenus *Kaempferia* with the exclusion of subgenus *Protanthium* was observed in previous molecular studies (*Techaprasan et al., 2010*; *Nopporncharoenkul et al., 2016*), including our results. Phylogenetic analysis reconstructed in the study was well-resolved and facilitated the refinement of the taxonomic classification of *Kaempferia* species. Based on our phylogenetic analysis, the genus *Kaempferia* was divided into three major distinct clades and generally congruent with the morphological pattern.

Subgenus *Kaempferia* includes clades A and B, while subgenus *Protanthium* is represented by clade C, based on the importance character of inflorescence appearance as mentioned above (Fig. 2). Firstly, clade A was split into four minor clades (A1–A4). In clade A1, the relationships of a complex species group of *K. galanga*, which consisted of taxa, namely *K. galanga*, *K. koratensis*, and an undescribed species (labelled as *K. chonburiensis*), were reported and clarified for the first time. This complex group was resolved, as evidenced by several morphological characters, including a large moniliform rhizomes, suborbicular leaves that are horizontal near the ground, and staminodes and labellum on the same plane (Fig. 3). However, the morphological characters in this complex group differ in ligule and peduncle length (more than 5 mm long in an undescribed species (labelled as *K. chonburiensis*) and less than 5 mm long in *K. galanga* and *K. koratensis*). From our morphological observations, the labellum color in this clade also has a white labellum with or without a violet patch. Moreover, *K. galanga* was believed to have been introduced from India and widely cultivated (*Holttum, 1950*; *Larsen & Larsen, 2006*), whereas *K. koratensis* (*Picheansoonthon, 2011*) and the undescribed species (labelled as *K. chonburiensis*) are endemic to Thailand. The difference between *K. koratensis* and the undescribed taxon was explained in diagnostic characters, compared in Table 1 and the character states are shown in Fig. 3. Accordingly, the undescribed

species mentioned as *K. chonburiensis* could be a single lineage separated from other closely related species, as evidenced by several autapomorphies.

Subsequently, the complex taxa, *K. larsenii* and *K. marginata*, in clade A2 were re-evaluated here, clarifying the polytomy found in the previous study (*Techaprasan et al., 2010*). The study resolved the close relationships in this complex between *K. larsenii* and *K. marginata*, and the phylogenetic position of *K. sisaketensis* was newly reported in this clade. Based on morphological evolution (Fig. 3), this clade A2 is closely related to clade A1 in ligule and peduncle length (less than 5 mm long) but differs in leaves orientation, staminodes, and labellum plane. Within this clade A2, the close relationships between *K. larsenii* and *K. marginata* agreed with the previous analysis (*Techaprasan et al., 2010*). According to the character tree as shown in Fig. 3, *K. larsenii* and *K. marginata* have moniliform rhizomes, whereas *K. sisaketensis* has cylindrical rhizome. However, *K. marginata* used in the study (Figs. 2, 3) was robustly supported by the molecular evidence and the character tree as an independent species. However, the previous interpretation of *K. marginata* was defined as a variation form of *K. galanga* proposed by *Jenjittikul, Nopporncharoenkul & Ruchisansakun (2023: 621)*. The position of *K. sisaketensis* was first reported as an ally with *K. larsenii* and *K. marginata*. In addition, the labellum color of members in this clade are violet. The staminode colors of *K. marginata* and *K. larsenii* are white and violet, respectively whereas the staminodes in *K. sisaketensis* are absent (*Picheansoonthon & Koonterm, 2009*). Furthermore, these three species are naturally distributed throughout northeast Thailand.

Clade A3 was closely related to clades A1 and A2 in rhizome pattern and ligule length (less than 5 mm long), and related to clade A1 in having staminodes and labellum on the same plane. Conversely, clade A1 to A3 differ in their floristic regions in Thailand. In clade A3, two related species, *K. pulchra* and *K. roscoeana*, were resolved and confirmed by their close relatives, according to *Techaprasan et al. (2010)*. Based on the character state analysis (Fig. 3), these two species are similar in having a moniliform rhizome, an absent to 5 mm long ligule, horizontal leaves, and staminodes and labellum on the same plane. However, they differ in the orientation of leaves and peduncle length. *K. roscoeana* has horizontal leaves appressed to the ground and a sessile to 5 mm long peduncle, while *K. pulchra's* leaves are not horizontal and its peduncle is longer than 5 mm. Subsequently, the flower color in *K. pulchra* is violet or white, whereas *K. roscoeana* is white. For their floristic regions in Thailand, *K. pulchra* is commonly found in southwest and peninsular regions, but *K. roscoeana* is distributed in north and southwest regions (*Jenjittikul, Nopporncharoenkul & Ruchisansakun, 2023*).

Later, the phylogenetic relationships between *K. elegans* and *K. parviflora* were placed in clade A4, which is consistent with the findings of *Techaprasan et al. (2010)*. Clade A4 is likely to be similar to clade A1 to A3 in rhizome pattern and ligule length (less than 5 mm long), but differs in peduncle length (more than 5 mm long), as shown in Fig. 3. Based on the character tree (Fig. 3), these two species are similar in having a moniliform rhizome, an absent to 5 mm long ligule, not horizontal leaves and an elongate peduncle. Nevertheless, they differ in that *K. elegans* has staminodes and labellum on the same plane and yellow

rhizomes, while *K. parviflora* does not have them on the same plane and dark violet rhizomes. This study also supported that *K. elegans* and *K. pulchra* were not combined as synonyms (*Searle, 1999*), so the clarification of two different species was clearly resolved following prior studies (*Sirirugsa, 1992*; *Larsen & Larsen, 2006*; *Techaprasan et al., 2010*).

Clade B is related to clade A1 to A4 in ligule length (less than 5 mm long) and to clade A2 in peduncle length (sessile to 5 mm). Conversely, it differs in the size and pattern of its rhizome, as shown in Fig. 3. Within clade B, the close relationships among three species from subgenus *Kaempferia*, *K. angustifolia*, *K. minuta*, and *K. fissa*, are observed in this study, as reported in the previous study (*Techaprasan et al., 2010*). Morphological observations of these species (Fig. 3) reveal similarities in their tiny rhizomes, sessile or very short ligules (up to 5 mm), and peduncles. However, there are some notable differences in their rhizome patterns, leaf orientation, and staminodes and labellum planes. *K. minuta* has a moniliform rhizome, while *K. angustifolia* and *K. fissa* do not have moniliform rhizomes. The leaves of *K. angustifolia* are elliptic, whereas those of *K. minuta* are elliptic and horizontal. In contrast, *K. fissa* has filiform leaves. The labellum is on the same plane in *K. minuta* and *K. fissa*, but not in *K. angustifolia*. According to their geological distribution, these species are recorded in northeastern Thailand and the adjacent country, Laos (*Jenjittikul, Nopporncharoenkul & Ruchisansakun, 2023*).

Finally, clade C includes *K. rotunda* and *K. udonensis*, which belong to the subgenus *Protanthium*. The monophyly of this subgenus is also confirmed in this study as same as the previous reports (*Techaprasan et al., 2010*; *Nopporncharoenkul et al., 2016*). These two species are characterized by precocious flowers as a key feature of the subgenus *Protanthium* (Fig. 3). Based on morphological evolution (Fig. 3), these two species are similar in having not moniliform rhizomes and sessile to 5 mm long peduncles. However, they differ in other features: *K. rotunda* has an absent or up to 5 mm long ligule, elliptic and not horizontal leaves, and staminodes and labellum that are not on the same plane. In contrast, *K. udonensis* has a longer than 5 mm ligule, suborbicular and horizontally oriented leaves appressed to the ground, and staminodes and labellum on the same plane. Moreover, *K. rotunda* is widely distributed from India to southeast Asia, while *K. udonensis* is endemic species to Thailand (*Phokham, Wongsuwan & Picheansoonthon, 2013*; *Jenjittikul, Nopporncharoenkul & Ruchisansakun, 2023*).

Consequently, our results differ from the traditional system proposed by *Horaninow (1862)* and *Baker (1890)* referring to morphological features only. The new infrageneric classification system is proposed here, and three infrageneric clades are supported by several substantial characteristics. Pending the results from our ongoing study, more extensive analysis of taxa in the genus is needed to refine the infrageneric classification of *Kaempferia* before a firm decision can be taken regarding the subgeneric circumscription.

## Phylogenetic placement of an undescribed taxon in *Kaempferia*

The systematic position of an undescribed taxon is investigated using the molecular phylogenetic approach based on ITS and *mat*K sequences to substantiate their differentiation. The combined taxonomic evidence, which includes morphological traits

(Figs. 4–6 and Table 1), molecular phylogenetic placement (Fig. 2) and morphological evolution *via* the character tree (Fig. 3), robustly supports the phylogenetic placement of the undescribed taxon (labelled as *K. chonburiensis*) as an evolutionarily distinct species within the genus *Kaempferia*. Despite several shared morphological traits with the closely related species *K. koratensis*, it exhibits consistent autapomorphies that distinguish it as a new species. Therefore, the newly identified species is circumscribed here as *K. chonburiensis*. A detailed description, along with illustrations (Fig. 4), photographs (Figs. 5, 6), a comparison with other related species (Table 1), and the key to species modified from the Flora of Thailand (Zingiberaceae) are provided in the section on Taxonomic treatment.

## Taxonomic treatment

*Kaempferia chonburiensis* Picheans., Sukrong & Wongsuwan, **sp. nov**. (Figs. 4–6)

**Type:** THAILAND. Chon Buri Province, Mueang District, at elevation of ca. 50 m, 24 June 2021, *PW 240621–1* (holotype: BKF; isotype: BK).

**Diagnosis:** This new species is similar to *Kaempferia koratensis*, but differs in the following characters: (1) longer ligule (0.6–1 cm *vs.* ca. 3 mm), (2) longer peduncle (0.8–1.2 cm long *vs.* ca. 5 mm long), (3) glabrous bract, (4) white with yellow spot at the base, with or without violet patch on labellum, (5) pubescent at the uppermost part of ovary.

**Description:** Perennial herbs. Rhizome bearing several tuberous roots. Bladeless sheaths 1–2, 2.8–5.2 cm long, pubescent. Leaf sheaths thick, 2.8–4.6 × 2.6–4.2 cm, green, pubescent; ligule 0.6–1.0 cm long, pubescent. Leaves usually two, sessile; blade horizontal near the ground, ovate to suborbicular, 14.6–23.9 × 16.4–24.1 cm, base rounded to cuneate, apex mucronate, margin red, upper surface dark green, glabrous, lower surface pale green, pubescent. Inflorescences enclosed in the two leaf sheaths; peduncle 0.8–1.2 cm, glabrous. Bracts lanceolate to oblong, 3.1–4.0 × 0.9–1.7 cm, apex acute, green with greenish to white at base, glabrous; bracteoles 2, linear, 2.9–3.7 cm × 2–3 mm, apex acute, translucent white, glabrous. Calyx tubular, 3.1–3.4 cm long, split on one side to 1.2–1.4 cm deep, apex bifid, white, glabrous. Corolla tube 3.4–4.7 cm long, white, glabrous; dorsal corolla lobe 1, oblong, 2.6–3.0 cm × 4–6 mm, apex hooded, white, glabrous; lateral corolla lobes 2, oblong, 2.0–2.4 cm × 3–5 mm, apex acute, white, glabrous. Staminodes 2, broadly obovate, 1.8–2.4 × 1.1–1.4 cm, apex rounded to slightly acute, white, glabrous. Labellum broadly obovate to suborbicular, bilobed, divided deeply to the base, each lobe obovate to broadly obovate, 2.6–3.1 × 1.5–2.1 cm, base clawed, apex undulated, rounded, white with yellow spot at the base, patch with or without violet. Anther 3–4 mm long; anther crest obovate, rectangular, 4–6 × 3–4 mm, apex rounded, bilobed. Stigma funnel-shaped. Ovary 4–5 × ca. 2 mm, pubescent at the uppermost part; stylodial glands 2, filiform, 4–6 mm long; ovule 3-locular, axile placentation. Fruit obovate, 0.9–1.5 × 0.6–1.0 cm, white. Seeds numerous, narrowly ellipsoid to deltoid, 3–4 × ca. 2 mm, aril white.

**Paratypes:** Thailand. Chon Buri Province, Mueang District, at elevation of 45–50 m, 12 July 2021, *PW 120721–1* (BKF); ibid., 10 June 2022, *PW 100622–1* (BKF).

**Phenology:** Flowering May-August; fruiting June-September.

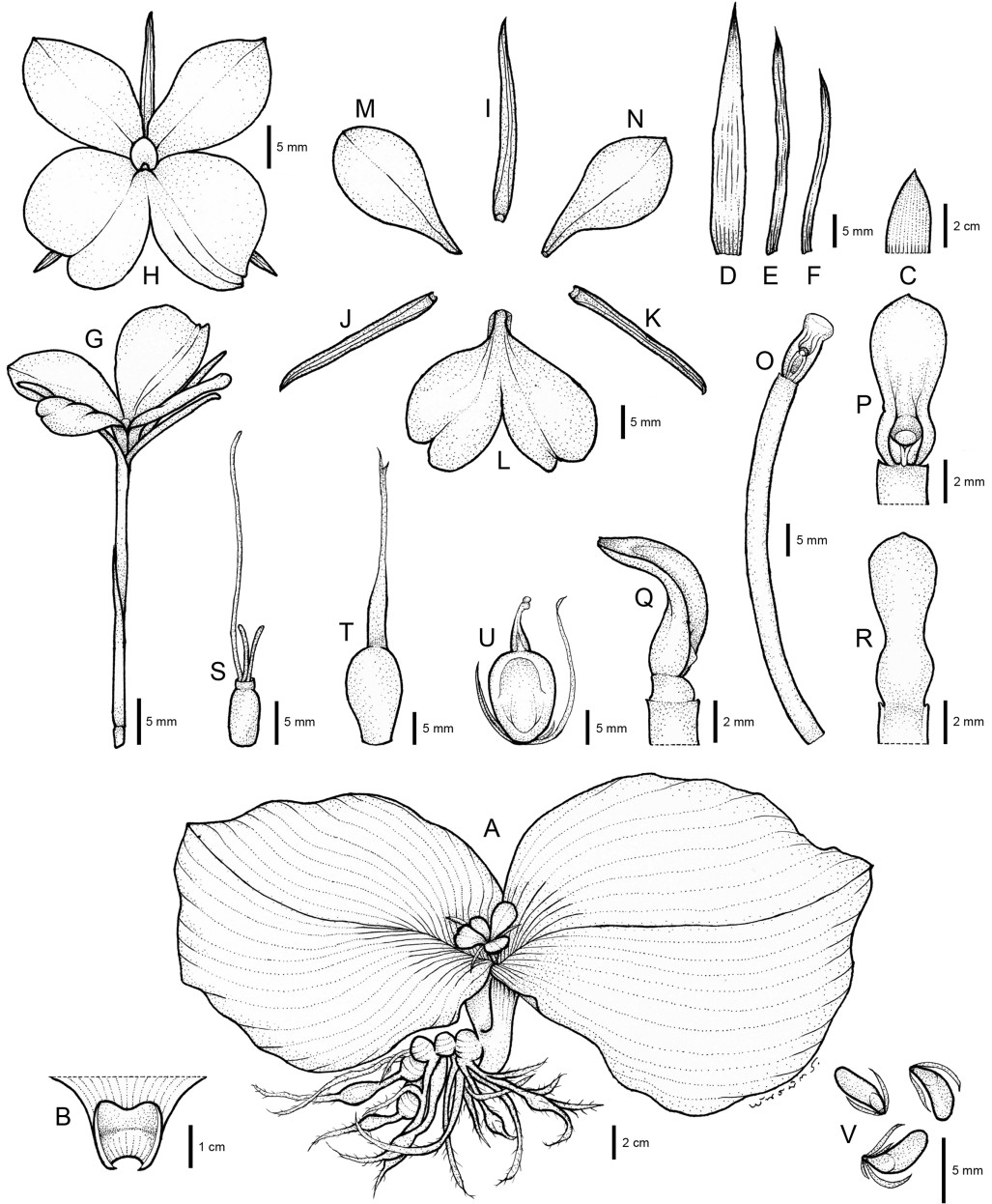

**Figure 4** **Drawing of *Kaempferia chonburiensis*.** (A) Whole plant. (B) Ligule. (C, D) Bracts. (E, F) Bracteoles. (G) Flower with calyx tube and ovary. (H) Flower (top view). (I) Dorsal corolla lobe. (J, K) Lateral corolla lobes. (L) Labellum. (M, N) Lateral staminodes. (O) Corolla tube with anther and anther crest. (P, Q, R) Anther and anther crest (front, side, and rear views). (S) Ovary, stylodial glands, and lower part of the style. (T) Immature fruit with persistent calyx. (U) Mature fruit with persistent calyx and bracteoles. (V) Seeds. Illustration based on the holotype (*PW 240621–1*) (drawn by Pantamith Ratta-nakrajang).

**Distribution:** This new species is known only from the type locality in Chon Buri Province, Thailand (Fig. 1).

**Ecology:** This new species grows under the shade of deciduous forest at elevations of 45–50 m.

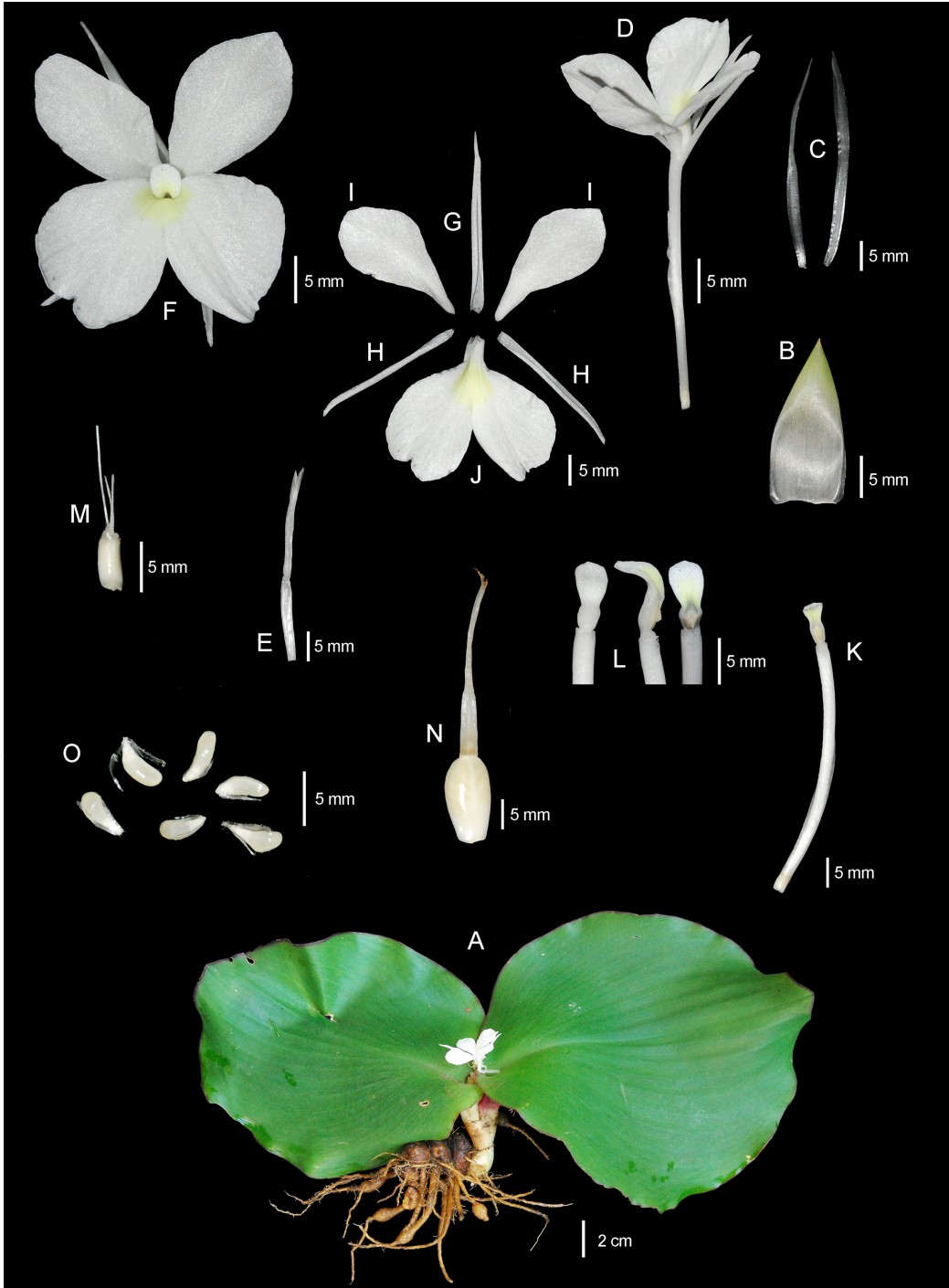

**Figure 5 The whole plant of *Kaempferia chonburiensis*.** (A) Whole plant. (B) Bract. (C) Bracteoles. (D) Flower with calyx tube and ovary. (E) Calyx tube. (F) Flower (top view). (G) Dorsal corolla lobe. (H) Lateral corolla lobes. (I) Lateral staminodes. (J) Labellum. (K) Corolla tube with ovary and anther. (L) Anthers and anther crests (front, side and rear views). (M) Ovary, stylodial glands, and lower part of the style. (N) Fruit and persistent calyx. (O) Seeds. Photographs based on the holotype (*PW 240621–1*) (photos by Pornpimon Wongsuwan).

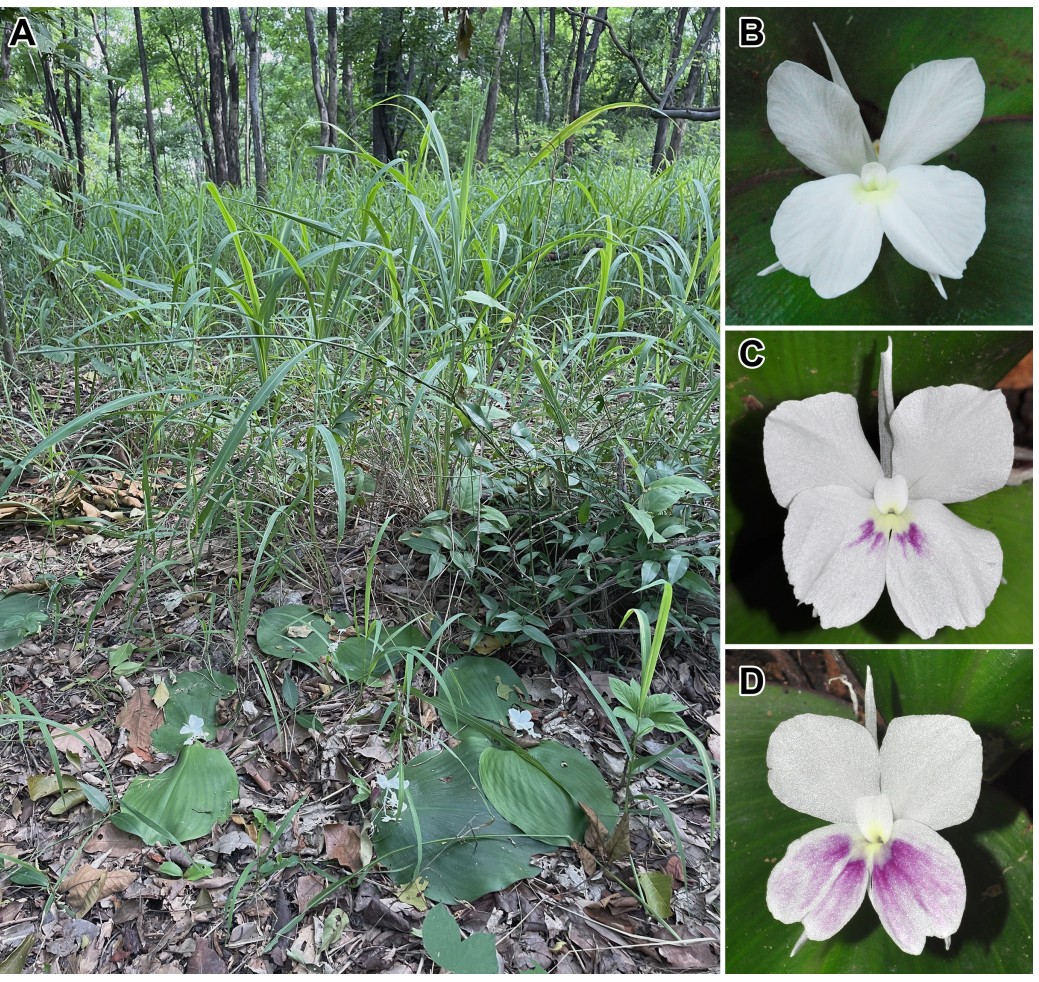

**Figure 6** *Kaempferia chonburiensis in situ.* (A) Natural habit in flowering stage. (B, C, D) Variation of flowers (labellum patch with or without violet). Photographs based on the holotype (A, B: *PW 240621–1*) and paratypes (C: *PW 120721–1* and D: *PW 100622–1*) (photos by Pornpimon Wongsuwan).

**Etymology:** The specific epithet 'chonburiensis' refers to the province of the type locality.

**Vernacular name:** Pro (เปราะ), Wan Pro (ว่านเปราะ), Pro Mueang Chon (เปราะเมืองชล).

**Uses:** Young leaves are used as vegetables and rhizomes as traditional medicines.

**Provisional IUCN conservation assessment:** This species is currently known only from the type locality, Mueang district, Chon Buri Province. While it is possible that the species also inhabits adjacent regions, our survey did not cover the entire area. Consequently, we propose classifying this species under the IUCN criteria (*International Union for Conservation of Nature (IUCN), 2022*) as data deficient (DD). We anticipate that further exploration of nearby unexplored areas may reveal additional distributions of this species, potentially leading to a reassessment of its conservation status and informing future conservation efforts.

**Identification key for *Kaempferia chonburiensis***

The new couplets to place *K. chonburiensis* would be inserted at couplet 26 in the key *Kaempferia* species (*Jenjittikul, Nopporncharoenkul & Ruchisansakun, 2023*: *612*).

1. Labellum white, patch with or without violet; anther crest ovate or obovate to rectangular.................................................................................... 2

2. Ligule shorter than 3 mm, labellum patch without violet. ................ *K. koratensis*

2. Ligule longer than 6 mm, labellum patch with or without violet...... *K. chonburiensis*

1. Labellum purple; anther crest quadrangular ............................... *K. chayanii*

## ACKNOWLEDGEMENTS

We thank Mr. Smittichai Sukplang for his assistance during fieldwork. Additionally, we are grateful to Miss Tawansap Apipobsombut for helping us prepare the figures. We acknowledge Sanith Sri Jayashan for language editing and suggestions to improve our manuscript.

### Funding

Pornpimon Wongsuwan's graduate program was supported the 90th Anniversary Chulalongkorn University Fund (Ratchadaphiseksomphot Endowment Fund). The funders had no role in study design, data collection and analysis, decision to publish, or preparation of the manuscript.

### Grant Disclosures

The following grant information was disclosed by the authors:
90th Anniversary Chulalongkorn University Fund (Ratchadaphiseksomphot Endowment Fund).

### Competing Interests

The authors declare that they have no competing interests.

### Author Contributions

- Pornpimon Wongsuwan conceived and designed the experiments, performed the experiments, analyzed the data, prepared figures and/or tables, authored or reviewed drafts of the article, and approved the final draft.
- Boonmee Phokham performed the experiments, analyzed the data, prepared figures and/or tables, authored or reviewed drafts of the article, and approved the final draft.
- Pantamith Rattanakrajang analyzed the data, prepared figures and/or tables, authored or reviewed drafts of the article, and approved the final draft.
- Chayan Picheansoonthon conceived and designed the experiments, authored or reviewed drafts of the article, and approved the final draft.
- Suchada Sukrong conceived and designed the experiments, authored or reviewed drafts of the article, and approved the final draft.

## DNA Deposition

The following information was supplied regarding the deposition of DNA sequences:

The ITS and matK sequences described here are available at GenBank: LC832953 to LC832969, and LC832970 to LC832986.

## Data Availability

Raw data are available in the Supplemental Files and at GenBank: LC832953 to LC832969, and LC832970 to LC832986.

The raw data shows voucher information, location, and GenBank accession numbers for samples used in this study.

## New Species Registration

The following information was supplied regarding the registration of a newly described species:

*Kaempferia chonburiensis* Picheans., Sukrong & Wongsuwan, sp. nov.; 77348702-1.

## Supplemental Information

Supplemental information for this article can be found online at http://dx.doi.org/10.7717/peerj.18948#supplemental-information.

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
