# Peer review of "Kaempferia chonburiensis (Zingiberaceae), a new species from Thailand based on morphological and molecular evidence"

_PeerJ, doi:10.7717/peerj.18948_

## Round 0.1 · original submission · Major Revisions

The reviewers have identified a need for various clarifications and some suggestions or concerns are identified, particularly related to Methods. Please carefully consider the reviewers' concerns when preparing a revised manuscript. Figure 6A is an excellent habitat photo, but this does not look like forest understory (the photo has grasses an no trees) whereas your species description says "grows under the shade of deciduous forest". Does this species grow in disturbed, open areas (as seemingly seen in Fig 6A) as well as intact forest understory. Can this point be clarified in the species description and / or figure 6 caption?

·

Basic reporting

Authors can enhance the clarity and impact of their articles by improving the language. Careful attention to word choice and sentence structure.

Experimental design

The experimental design for the study is fine and thoughtfully conceived. It effectively addresses the research questions and provides a solid framework for obtaining reliable and meaningful results.

Validity of the findings

no comments

Additional comments

The article titled "Kaempferia chonburiensis (Zingiberaceae), a New Species from Thailand Based on Morphological Evidence and Phylogenetic Analysis" presents a newly identified species, Kaempferia chonburiensis, from the genus Kaempferia. The authors have provided substantial morphological and molecular evidence to support this claim. The manuscript is well-structured. However, there are several major aspects that need to be addressed before it can be accepted for publication.

In the abstract, the author mentions that there are 60 accepted species in Kaempferia. However, this number appears to differ from the current accepted count. I recommend verifying this information for accuracy.
Line 33- Bayesian Inference
Correct the punctuations at line no, 36, 109, 126, and 199
Line 57- the number provided differs from that mentioned in the abstract. Please correct one of them to ensure consistency throughout the manuscript.
Line 92- other
Why did the author decide to choose ITS and matK markers for phylogenetic studies? What was the thought behind it? As matK is a coding conserved gene, please, clarify its use.
The author combined nuclear and chloroplast regions for phylogenetic analysis. Did the author attempt to analyze the nuclear and plastid datasets separately to identify any significant differences between them? Additionally, did they perform a data incongruence test, such as the ILD test, before combining these datasets to ensure compatibility?
Line 131 to 134- move to last para of same section.
Line 140- included/ used
I suggest to tabulate the primer sequences and PCR reaction conditions
Line 159- where is appendix A?
Which model used for ML analysis?
Line 175- in table 2
Line 185- belong
Line 193- are indicated
The author selected only one outgroup for the analysis. However, when assessing monophyly, a single outgroup may not provide sufficient context. I recommend including additional outgroups, either from the same outgroup species or from closely related species, to strengthen the support for monophyly in the phylogenetic analysis.
Line 196- remove as revealed by the maximum bootstrap
The author notes that in Kaempferia chonburiensis, some flowers exhibit a violet patch while others do not. Could this indicate a potential complex within K. chonburiensis? It would be an intriguing area for further study if the author could investigate whether this variation represents intraspecific diversity or suggests a cryptic species complex.
Line 232- author should test monophylyl with more outgroups
Line 241- remove always
Line 270- was/ is
Line 292- …..while K. pulchra’s leaves are not horizontal and a peduncle is longer……..
Line 302-304- rewrite it
Line 319- start with “According to their…
In Figure 2, the node support values are difficult to interpret. I suggest displaying the node values, specifically the maximum likelihood bootstrap values (BS > 50%) and Bayesian inference posterior probabilities (PP > 0.50), for clearer understanding. If the major clades still exhibit low node support, I recommend including another plastid spacer region in the analysis alongside matK to enhance the robustness of the phylogenetic findings.

Reviewer 2 ·

Basic reporting

This is a well-structured paper. The authors describe a new species, Kaempferia chonburiensis, from Thailand based on morphological evidence and phylogenetic analysis using ITS and matK. The descriptions are very thorough and well-written, including all necessary information about the new species. The illustrations seem fine, and the phylogenetic tree is well-structured. Overall, it is a thorough job. However, I have a few comments.
1. Line 109: The names of the herbaria should be mentioned and not acronyms (BK, BKF, QBG, and SING).
2. Line 126: Since the genus has close to 60 species, 32 of which are found in Thailand, I was expecting the authors to use at least 70 percent of the species.
3. Line 162-163: Maximum Likelihood (ML) analysis was
chosen to perform on XSEDE using IQ-Tree on ACCESS (2.2.2.7). This should read, "Maximum Likelihood analysis was performed using IQ-Tree (XSEDE, on ACCESS (2.2.2.7))."
4. 163: 1,000 bootstrap should read "1,000 ultra-fast bootstrap "
5. Please use the recent version of FigTree v.1.4.4 and note the same for other software used. Also, please mention the version of the packages used, for instance, phytools v.2.0 (Revell, 2024).

Table 1: The authors should consult relevant literature for the elevation of K. galanga and K. roscoeana. For instance, the elevation of K. galanga can be found in the work of Preetha et al. (2016).

Preetha TS, Hemanthakumar AS, Krishnan PN. A comprehensive review of Kaempferia galanga L. (Zingiberaceae): A high sought medicinal plant in Tropical Asia. J. Med. Plants Stud. 2016;4(3):270-6.

Experimental design

The submission describes the research well within the scope of the Journal. The research question was clearly defined. The investigation has been well conducted and deemed acceptable. Methods are described with sufficient information, are straightforward, and are reproducible.

Validity of the findings

The findings are consistent with the research question, sequence data deposited with accession numbers and herbarium specimen with voucher numbers. The conclusions are not specifically included as a section but embedded and can be inferred from the Results, they are connected to the original question investigated and limited to those supported by them.

Reviewer 3 ·

Basic reporting

This manuscript describes a new species of Kaempferia based on morphological and molecular evidence. This manuscript includes good sampling, molecular, and morphological data, which could help to clarify the taxonomy incongruences within the group. The manuscript was prepared in good English. However, there are some methodological issues that could be revised. Also, some methodological aspects are missing in the text, which could compromise the understanding and the replicability of the analysis (easily fixed by adding the missing information).
Although the manuscript has some aspects in need of careful revision, this is a very interesting model of study. So, I really encourage the researchers to review some aspects and re-submit the manuscript.

Experimental design

no comment

Validity of the findings

no comment

Additional comments

no comment

Annotated reviews are not available for download in order to protect the identity of reviewers who chose to remain anonymous.

---

## Round 0.2 · Minor Revisions

The reviewer appreciated revisions that were made and has provided an additional comment that should be address in your revision and/or rebuttal. Additionally, I provide the following suggested minor edits to the text:
L 30 “The phylogenetic trees” to “Phylogenetic trees”
L 31 “The morphological evolution” to “Morphological evolution”
L 88 “plant” to “plants”
L 91 “the phylogenetic reconstruction” to “phylogenetic reconstruction”
L 93 “provide key” to “provide a key”
L 112 “Distribution map” to “A distribution map”
L 172 “Seven characters of morphological characteristics” to “Seven characters based on morphology”
L 213 “character is composed of having inflorescence” to “character is the inflorescence”
L 233 “the color changes whether this variation corresponds” to “whether color differences correspond”
L 235 “help more understanding the” to “help improve understanding of the”
L 257 “was recognized into” to “was split into”
L 267 “widely cultivated species” to “widely cultivated”
L 269 “are endemic” to “is endemic”
L 277 “Based on the” to “Based on”
L 282 “were moniliform” to “have moniliform”
L283 “was cylindrical” to “has cylindrical”
L 284 “independence” to “independent”
L 285 “its controversial species” – Which species? Please specify here.
L 285-287 “whereas K. galanga is another species, differed from the descriptive notes of K. galanga proposed in Jenjittikul, Nopporncharoenkul & Ruchisansakun (2023: 621)” - I did not understand this claim – is this claim based on your DNA evidence or morphological evidence? Then what is the species in the descriptive notes? This is confusing as currently worded and needs further explanation; or, if it is not a crucial point it could be removed from the manuscript.
L 313 “as the synonym” to “as synonyms”
L 323 “leaves orientation” to “leaf orientation”
L 334 “the morphological” to “morphological”
L 346 “more extensive taxon” to “more extensive analysis of taxa”
L 358 “distinguish them” to “distinguish it”

Reviewer 3 ·

Basic reporting

The revision was prepared carefully to address the questions that were highlighted.
However, there are some minor questions that need to be resolved before the manuscript can be accepted for publication.

The JModelTest can only provide you the best-fitting model for the data set to build the phylogenetic tree. It can not show you incongruence between the data sets.
I suggest you mention the congruence between nuclear and chloroplast data in the "Molecular phylogenetic analysis" of the "Results" section.

For PP support value use 1.0

Experimental design

no comment

Validity of the findings

no comment

Additional comments

no comment

---

## Round 0.3 · accepted · Accept

Thank you for submitting minor revisions to your manuscript.

I suggest only two minor English edits to the abstract:
1. Rather than "comprised", change to "is comprised of"
2. Rather than "from the fields", change to "from field sites"